# Tackling Acute Lymphoblastic Leukemia—One Fish at a Time

**DOI:** 10.3390/ijms20215313

**Published:** 2019-10-25

**Authors:** Arpan A. Sinha, Gilseung Park, J. Kimble Frazer

**Affiliations:** 1Jimmy Everest Section of Pediatric Hematology-Oncology, Department of Pediatrics, University of Oklahoma Health Sciences Center, Oklahoma City, OK 73104, USA; Arpan-Sinha@ouhsc.edu (A.A.S.); Gilseung-Park@ouhsc.edu (G.P.); 2Department of Cell Biology, University of Oklahoma Health Sciences Center, Oklahoma City, OK 73104, USA

**Keywords:** leukemia, B-ALL, T-ALL, zebrafish, MYC, leukemia models

## Abstract

Despite advancements in the diagnosis and treatment of acute lymphoblastic leukemia (ALL), a need for improved strategies to decrease morbidity and improve cure rates in relapsed/refractory ALL still exists. Such approaches include the identification and implementation of novel targeted combination regimens, and more precise upfront patient risk stratification to guide therapy. New curative strategies rely on an understanding of the pathobiology that derives from systematically dissecting each cancer’s genetic and molecular landscape. Zebrafish models provide a powerful system to simulate human diseases, including leukemias and ALL specifically. They are also an invaluable tool for genetic manipulation, in vivo studies, and drug discovery. Here, we highlight and summarize contributions made by several zebrafish T-ALL models and newer zebrafish B-ALL models in translating the underlying genetic and molecular mechanisms operative in ALL, and also highlight their potential utility for drug discovery. These models have laid the groundwork for increasing our understanding of the molecular basis of ALL to further translational and clinical research endeavors that seek to improve outcomes in this important cancer.

## 1. Introduction

The word ”Leukemia” was coined from German, and derives from the Greek words “*leukos*”, meaning white + ”haima”, meaning blood. Leukemias are cancers of white blood cells, and based on their cell of origin, are divided into leukemias of lymphoid/lymphocytic or myeloid/myelocytic origin. Based on their rate of clinical progression, they are categorized as either acute or chronic. This schema results in four broad leukemia classifications—acute lymphoid/lymphoblastic leukemia (ALL); acute myeloid/myelogenous leukemia (AML); chronic lymphocytic leukemia (CLL); and chronic myeloid/myelocytic leukemia (CML). These designations predict the clinical course and guide treatment. This review focuses exclusively on ALL.

An estimated ~6000 total new cases of pediatric and adult ALL occur in the U.S. annually, accounting for ~1500 deaths/year with most ALL deaths (about 80%) occurring in adults [1]. In childhood, ALL is arguably the most important cancer. ALL is the most common pediatric malignancy in the U.S., representing >25% of all cancer in children 0–14 years [2,3]. About 85% of cases are precursor-B (pre-B ALL); the remainder are T-ALL [3,4]. Despite huge strides in pediatric ALL treatment, ALL is still the second most lethal childhood cancer, causing ~25% of deaths [5]. Pediatric ALL is so common, relapsed ALL is actually the fourth most-frequent pediatric cancer diagnosis, with pre-B ALL the main entity [6]. Historically, T-ALL has been considered more aggressive, requires more intense therapy, has inferior outcomes after relapse, and has higher chances of CNS involvement and CNS relapse [7], but evolving treatment paradigms have improved pediatric T-ALL outcomes with respect to pre-B ALL [8]. This review focuses on zebrafish ALL models. Traditionally, ALL is categorized as precursor-T (or T-cell (T-ALL)), precursor-B (or pre-B cell (B-ALL)), or mature B cell (Burkitt ALL) based on immunophenotypes of distinct lineage and cell maturational markers. These can be further subdivided according to recurrent karyotypic abnormalities, including specific aneuploidies and translocations [9,10]. Although grouped together as ALL, B-ALL and T-ALL exhibit unique biologies that underlie their differing clinical behavior and drug sensitivities. Within each category, multiple genetic sub-categories exist, and these demonstrate even further heterogeneity. Defining the specific features of each genetic subtype of ALL is an active area of investigation that will, hopefully, allow oncologists to tailor therapies matching each ALL’s molecular anomalies, a concept oft-termed ”precision oncology”. Zebrafish can help with this, because their genetic manipulability is unmatched among vertebrate models, and they are already proven to be capable of developing various ALL types that emulate human ALL. Genomic profiling and mutational analyses are refining ALL classification at a rapid pace, identifying new genetic lesions that coexist and cooperate with other known alterations, and finding new ALL subtypes in cases lacking known cytogenetic alterations [10]. Making sense of the discoveries gleaned from human clinical samples at a molecular and mechanistic level is likely one of the principal ways that zebrafish models can enhance progress.

### Zebrafish are a Model Befitting Leukemia Modeling

Over the past few decades, zebrafish (*Danio rerio)* have emerged as a powerful vertebrate model to recapitulate many diseases, including cancer. There are several attributes that favor *D. rerio* relative to other vertebrates: Low cost of housing, largely due to their small size; high fecundity; external fertilization, ex-utero development, and translucent embryos, all enabling real-time visualization of early embryo- and organo-genesis; ease and speed of genetic manipulation, including simple transgenesis and precise genome-editing approaches to knock-down and overexpress genes by various technologies; and capacity for phenotype-driven genetic and chemical screens to identify genetic interactions, new molecular targets, novel agents, or to re-purpose existing drugs for new indications.

Given their distant evolutionary relationship, it is perhaps surprising how similar zebrafish and humans are genetically. Howe et al. showed that ~70% of human genes have zebrafish orthologues [11]; this aids in correlating gene expression in zebrafish cancers to their human counterparts and with designing transgenic zebrafish that express human (or other mammalian) genes. *D. rerio* have repeatedly proven to be an excellent model to study blood disorders and hematologic cancers [12,13,14,15,16,17,18,19]. The zebrafish adaptive immune system resembles that of humans [20], and zebrafish thymic architecture is also similar to mammals, containing distinct cortical and medullary regions that compartmentalize maturing T cells [21]. In addition, the genetic programs governing hematopoiesis, oncogenesis, and tumor suppression are highly conserved between these species [22,23], so much so that transgenic mammalian proteins typically show conserved function in zebrafish. Transgenic fluorescent reporter lines have also been established, enabling live in vivo imaging, cell-lineage tracing, or rapid microscopy screening for cancers. Transplantation protocols, and even immunodeficient zebrafish lines, are now available [24,25,26,27,28], further accelerating *D. rerio* leukemia studies. Several groups have exploited these and other tools to study ALL in zebrafish. Below, we summarize key studies that have advanced our understanding of human ALL employing experimental strategies in *D. rerio*.

## 2. Zebrafish Acute Lymphoblastic Leukemia Models

### 2.1. T-ALL: Introduction

T-ALL is an aggressive hematologic malignancy characterized by uncontrolled division of immature T-progenitors (i.e., T-lymphoblasts). T-ALL represents ~25% of adult and ~15% of pediatric ALL cases [3,29]. Common clinical presentations include hyperleukocytosis and extramedullary involvement of lymph nodes and/or other organs such as the central nervous system, spleen, and mediastinal masses, which arise in the thymus [30]. T-ALL and T-cell lymphoblastic lymphoma (T-LBL) are often considered as one disease, differing only by their extent of bone marrow infiltration, and treatments for both are similar, if not identical [31]. Zebrafish models of T-ALL/LBL have advanced dramatically since the groundbreaking work of Langenau et al., who described the first zebrafish T-ALL model in 2003 [32]. Not only was this the first transgenic ALL model in *D. rerio*, but also the first zebrafish cancer model overall. Now, the number of zebrafish T-ALL models has reached double digits, as summarized below [12,13,14,15] (Table 1).

### 2.2. Zebrafish T-ALL Models

#### 2.2.1. Model 1: T-ALL Induced by Murine *Myc – Tg*(*zrag2:EGFP-mMyc*) Zebrafish

The first demonstration of stable leukemia-prone transgenic fish used a construct consisting of a zebrafish lymphoblast-specific *recombination-activating gene-2* (*zrag2*) promoter driving a murine *c-Myc (mMyc)* oncogene. The transgene also contained enhanced green fluorescent protein (EGFP); thus, cancers were fluorescent and could be detected and monitored by simple microscopy [32]. To create this line, fish were microinjected with the transgene as one-cell embryos. All mosaic fish that integrated the transgene developed T-ALL, proving high (100%) penetrance of the phenotype. RNA in situ hybridization confirmed *mMyc* expression, as well as that of T-lineage genes, proving leukemias were T-ALL. Flow cytometric DNA analyses demonstrated T-ALL were clonally aneuploid. T-ALL ”immortality” was shown by transplantation into irradiated recipients, with rapid engraftment seen [32]. Leukemias resembled human T-ALL in many ways—they grew rapidly after very short latency that averaged only 52 days post-fertilization (dpf), invaded into multiple lymphoid and non-lymphoid tissues, and exhibited clonal TCR-α gene rearrangements.

Complete penetrance and rapid leukemic onset represented a huge scientific success, but it also created a challenge: T-ALL grew so quickly that many animals died before sexually mature. Thus, maintaining this line was difficult. Ensuing derivative transgenic lines (detailed below) rectified this. Alternatively, in vitro fertilization (IVF) of preserved sperm can maintain *zrag2:mMyc* fish, or other strategies can be employed: First, since *zrag2:mMyc* is so potent, many labs inject the construct into new embryo clutches to make T-ALL as-needed in whatever genetic background they desire. Second, by freezing T-ALL cells, leukemias can later be injected into new recipients (or even passaged serially through multiple transplant rounds), to propagate individual T-ALL indefinitely.

##### Key Discoveries

• Role of MYC in ALL

The success of this model reinforced that MYC was a powerful oncogene, potently driving leukemogenesis across species. MYC’s key role in T-ALL and other hematologic malignancies was already established [33,34,35,36,37], and studies of MYC in ALL continue to be published regularly [38,39]. In fact, MYC was also found to drive zebrafish B-ALL, as discussed subsequently.

• *mMyc* T-ALL Mirrors Most Common Human T-ALL subtype

Characterization by RT-PCR and in situ hybridization showed cancers express both *tal1*/*scl* and *lmo2* (fish orthologues of known human T-ALL oncogenes *TAL1/SCL* and *LMO2*), emulating the most common molecular T-ALL subtype in patients [40,41]. Of note, the *zrag2* promoter used in this line has been utilized to create nearly every other transgenic zebrafish ALL model thus far described, attesting to its utility, but also revealing a limitation in the field that investigators should address in future studies.

• Role of DLST and TCA Metabolism in T-ALL

Others have used “*mMyc* fish” to find T-ALL genetic modifiers. Anderson et al. performed a candidate genetic screen that revealed a role for dihydrolipoamide S-succinyltransferase (DLST) and tricarboxylic acid cycle (TCA) metabolism in T-ALL [42]. DLST is part of a TCA cycle enzymatic complex that dehydrogenates α-ketoglutarate (α-KG) into succinyl-CoA, and heterozygous inactivation of *dlst* significantly delayed *mMyc*-driven zebrafish T-ALL. Findings were corroborated in human T-ALL cell lines, where *DLST* RNAi knockdown led to impaired viability and apoptosis due to TCA cycle disruption and α-KG accumulation. Addition of the downstream intermediate succinate rescued DLST inactivation-induced viability defects. Overall, this study proved metabolic dependence of T-lymphoblasts on the TCA cycle, providing opportunities for targeted therapy [42].

• Creating a Transplantable Zebrafish T-ALL Cell Line with *zrag2:EGFP-mMyc* CG2 Fish

Mizgirev et al. created a zebrafish T-ALL line, ZL1, to screen anti-leukemia agents. Specifically, they built a T-ALL line that could be transplanted into syngeneic hosts, facilitating production of many zebrafish bearing synchronously growing T-ALL [43]. They first created a novel CG2 (Clonal Golden 2) line comprised of genetically near-identical fish [44,45]. They then injected *zrag2:EGFP-mMyc* into CG2 embryos to create the ZL1 T-ALL line, and subsequently propagated it through >20 transplant passages in adult CG2 fish. Next, they tested the human T-ALL chemotherapeutics cyclophosphamide (CY), vincristine (VCR), and prednisolone (PRE) for efficacy in zebrafish larvae harboring ZL1, using increased lifespan as their outcome variable. They demonstrated dose- and time-dependent effects for CY and VCR, but as in human T-ALL, monotherapy with either did not cure disease. In their study, PRE was not effective at doses of 1–50 mg/L for treatments of 24–72 h due to toxicity occurring exclusively in ZL1-bearing larvae. This model, or others like it, could provide cost-effective platforms for large-scale anti-cancer compound screens [43].

#### 2.2.2. Model 2: T-ALL in Conditional *mMyc* Models

##### *Model 2a: rag2:loxP-dsRED2-loxP-EGFP-mMyc [*aka, *Tg(rag2:LDL-EMyc)] Zebrafish*

The *mMyc* line was modified, using conditional *Cre-Lox* technology to overcome the need for IVF [40]. Here, the *EGFP-mMyc* construct was preceded by a *loxP-DsRed2-loxP* cassette. In theory, without Cre, *mMyc* would not be expressed and *dsRED2* (a red fluorophore) would label immature thymocytes. To express *EGFP-mMyc*, *Cre* recombinase could be injected into one-cell embryos carrying the transgene, causing *loxP*-mediated excision of *dsRED2* and expression of *EGFP-mMyc*. Presumably due to less-than-100% Cre efficiency, fish developed T-ALL with longer mean latency of 151 dpf (vs. 52 dpf in the original line), and T-ALL remained morphologically similar to the original *mMyc* line. However, suboptimal recombination and mosaic *EGFP-mMyc* activation did pose challenges. Only ~7% of Cre-injected fish developed T-ALL, and most expressed both dsRED2 and GFP, suggesting leukemic cells had only partial Cre recombination of a transgenic locus harboring multiple *rag2:loxP-dsRED2-loxP-EGFP-mMyc* copies [40]. Mosaicism was further indicated by a single fish with one T-ALL clone expressing both dsRED2 and GFP in one thymus, and a second T-ALL expressing only GFP in the other thymus [40].

##### Model 2b: Double-Transgenic [*Tg*(*hsp70:Cre;rag2:LDL-EMyc*)] Zebrafish

A variant of the conditional model above improved its low penetrance. Here, Cre injection was replaced by a heat shock-inducible *Cre* transgene, *hsp70:Cr*e [46]. Double-transgenic *(hsp70:Cre;rag2:LDL-EMyc)* 3 dpf embryos were heat shocked to induce Cre. This markedly improved T-ALL penetrance, (~80%), with mean latency of 120 dpf, overcoming both limitations of prior models. These fish were easier to maintain, and amenable to studies like forward-genetic screens. A (potential) drawback of this line was that without heat shock, T-ALL still occurred in 13% of animals, although ‘leakiness’ was lessened by raising fish at 24 °C and/or PCR-screening for fish with no or little recombination. Even so, ~90% of non-heat shocked fish remained cancer-free by 201 dpf, so these fish were simpler to maintain [46].

• Key Discoveries

Using this Cre-inducible line, the original *rag2:mMyc* line, and a third transgenic with human *MYC* (detailed subsequently), Huiting et al. investigated ubiquitin fusion degradation 1 (UFD1) in *mMyc*-driven zebrafish T-ALL [47]. They found MYC up-regulated UFD1 in T-ALL, and allelic loss of zebrafish *ufd1* induced T-ALL apoptosis, impairing cancer progression. Akin to the previously cited DLST study, this identified UFD1 as a genetic modifier in MYC-driven T-ALL, with potential utility for developing targeted therapy [47].

##### Model 2c: Triple-Transgenic [*Tg*(*hsp70:Cre;rag2:LDL-EMyc;rag2:EGFP-bcl2*)] Zebrafish

Double-transgenic heat shock-inducible *mMyc* fish were further modified to analyze the role of zebrafish *bcl2*, a BH3-family anti-apoptotic gene, with respect to autophagy and thymic T-LBL progression to disseminated T-ALL. This is one of the only studies to probe differences between these two closely related cancers often viewed as one disease [48]. Triple-transgenic fish with *hsp70:Cre*, *rag2-LDL-EGFP-mMyc*, and *rag2:EGFP-bcl2* were built, creating T-lymphoblast cancers with constitutive *bcl2* overexpression. They exhibited earlier onset and higher penetrance of T-LBL than double-transgenic fish lacking *rag2:bcl2*, suggesting *bcl2* accelerated MYC-induced T-cell cancer.

• Key Discoveries

Remarkably and likely unexpectedly, while accelerating T-LBL, *bcl2* overexpression profoundly inhibited T-LBL progression to T-ALL. Cells overexpressing *bcl2* failed to invade the vasculature and showed increased autophagy. This prompted the conclusion that autophagy was due to the inability of T-LBL cells to disseminate—they proliferated until they exhausted local nutrient supply, causing metabolic stress and autophagy. Conversely, lower Bcl2 levels allowed alternative cell survival programs through pathways promoting dissemination to avoid autophagy. This potentially unravels a biological basis to explain why some patients have localized T-LBL, while others exhibit rapidly disseminating T-ALL [48]. In addition, Feng et al. showed elevated S1P1 signaling and increases in its downstream target ICAM1 promoted homotypic cell adhesion through ICAM1–LFA1 binding. Together, this impeded intravasation and thymic egress, preventing hematologic dissemination and promoting autophagy. Their findings were confirmed in human biopsies, where autophagy markers and increased BCL2α, S1P1, and ICAM1 were seen in T-LBL compared to T-ALL. S1P1 inhibition in T-LBL cells decreased homotypic adhesion in vitro and increased tumor intravasation in vivo. An important potentially translatable finding from this study was that AKT promoted T-LBL progression to T-ALL, suggesting PI3K-AKT inhibitors might prevent T-LBL dissemination [48].

#### 2.2.3. Model 3: Co-Injection Strategies Pairing *zrag2:EGFP-mMyc* with Other Transgenes

Langenau et al. reported another approach to modify T-ALL initiation, co-injecting multiple transgenes to overcome some limitations in the aforementioned models [49]. Their method aimed to rapidly assess synergism or antagonism between combinations of genes, by injecting multiple transgenes simultaneously. To assess whether different transgene constructs co-integrated, *rag2:mMyc* and *rag2:dsRED2* were co-injected into stably transgenic *rag2:GFP* fish. All T-ALL that developed were both GFP+ and dsRED2+, indicating transgenes co-integrated and were co-expressed. With the knowledge that multiple transgenic constructs behaved as one in vivo, this study provided key proof-of-concept that co-injection strategies could rapidly interrogate (potentially) collaborating genes and pathways [49].

##### Key Discoveries

Co-injection is powerful because hundreds of embryos can be injected with multiple transgene combinations; consequently, many investigators now employ this. Examples utilizing this approach include the following:

• Transplantation into Syngeneic CG1-Strain Zebrafish

Smith et al. co-injected one of four fluorescent transgenes (rag2:GFP, *rag2:dsRED*, *rag2:zsYellow*, or *rag2:Amcyan*) plus rag2:mMyc to create different colored T-ALL in the syngeneic CG1 background [44,45]. They then transplanted very low numbers of T-ALL cells into CG1 recipients, which allowed transplantation into hosts without irradiation or immunosuppression [44]. T-ALL successfully engrafted from very few cells, accurately quantifying leukemia-initiating cells (LIC) and showing that 0.1–1.4% of *mMyc* primary T-ALL cells have LIC activity. Using large-scale single-cell transplantation, they further proved T-ALL can arise from single transplanted cells, and that LIC frequencies vary considerably between individual T-ALL. Remarkably, they transplanted >1400 fish in this study, showing that zebrafish are amenable to large-scale transplant projects that would be financially and practically infeasible in mammalian models [44].

• Modeling Treatment Resistance and Relapse

Similarly, co-injecting *m*Myc** together with any of several fluorophores, Blackburn et al. investigated how clonal evolution and heterogeneity drive cancer progression, studying differences between T-ALL clones [50]. Functional variation in individual clones revealed changing growth rates and LIC frequencies over time. Some T-ALL cells spontaneously acquired mTORC1 activation, which stabilized MYC to shorten T-ALL latency and increase LIC. This also fostered dexamethasone resistance, despite no prior exposure. AKT inhibition reversed this. They concluded T-ALL cells stochastically acquire mutations that cause treatment resistance even before drug exposure, in effect driving relapse before therapy even begins. Further, in their model, dexamethasone and AKT inhibition showed anti-LIC activity [50].

#### 2.2.4. Model 4: Inducible T-ALL in *rag2:hMYC-ER* Transgenic Zebrafish

MYC is an established oncogenic transcription factor that is highly expressed by many T-ALL, and NOTCH1 activation, also common in T-ALL, fosters MYC expression [34,35,36]. The PTEN-PI3K-AKT axis is also frequently deranged in T-ALL [51,52], and cases with *PTEN* and/or *PI3K-AKT* alterations often overexpress *MYC* too [51]. To explore these interactions, Gutierrez et al. built a conditional zebrafish model with transgenic human *MYC* (*hMYC)*. They fused *hMYC* to a modified estrogen receptor (*ER*) that binds 4-hydroxytamoxifen (4HT), but not endogenous estrogens. Once again, the zebrafish *rag2* promoter controlled transgene expression. 4HT treatment causes nuclear translocation of the *hMYC-ER* fusion, thereby post-translationally activating MYC in an inducible, and reversible, manner. Exposing *rag2:hMYC-ER* larvae to 4HT beginning at 5 dpf caused 100% T-ALL penetrance by five weeks of age, but many cancers regressed upon 4HT withdrawal, suggesting some ‘leukemias’ were not stably transformed, but rather lymphoproliferations that relied upon very high MYC activity. Subsequent studies revealed even without 4HT treatment, T-ALL occurs in most animals by nine months [53,54].

##### Key Discoveries

• MYC, PTEN, and PI3K-AKT Interactions in T-ALL

This study demonstrated loss-of-function of endogenous *pten,* or a constitutively active myristoylated murine *Akt2* (*myr-mAkt2*) transgene, both render T-ALL less reliant upon MYC, promoting T-ALL persistence despite 4HT withdrawal. This suggests MYC-dependency is partly mediated by Pten, which inhibits the Pi3k-Akt pathway. Additional studies indicated Akt promoted T-cell migration, suppressed autophagy, and inhibited apoptosis [55]. Collectively, their findings implied PI3K-AKT inhibitors may be useful—especially if combined with MYC inhibitors—for T-ALL treatment. Reynolds et al. further explored these ideas, investigating the pro-apoptotic protein BIM with respect to MYC. Using *rag2:hMYC-ER* fish, they showed MYC down-regulation induced BIM, favoring apoptosis, but this was blunted by constitutively active *myr-mAkt2* [56]. They extended this concept to human T-ALL studies, concluding that BIM repression is a key event downstream of MYC and PI3K-AKT in treatment-resistant T-ALL.

• Drug Screening in *rag2:hMYC-ER* Zebrafish

*hMYC* fish were also used in a drug screen seeking agents cytotoxic to MYC-overexpressing thymocytes [57]. Embryos carrying a *rag2:dsRed2* marker, either with or without *rag2:hMYC-ER*, were incubated with 4HT to activate MYC-ER, plus 4880 agents from four compound libraries. After four days (3–7 dpf), thymic fluorescence was assessed. Ultimately, this work revealed Perphenazine (PPZ), an FDA-approved antipsychotic, could induce apoptosis of fish, mouse, and human T-ALL. They identified protein phosphatase 2A (PP2A) as a PPZ target that was activated—not inhibited—by PPZ. Treatment with PPZ suppressed human T-ALL growth, dephosphorylating PP2A targets in vitro and in vivo. Overall, this suggested therapeutic potential for PP2A as a target in T-ALL [57].

• Discovery of T-ALL in *rag2:myr-mAkt2* Zebrafish

The original *hMYC* report also made a second, somewhat underappreciated finding. In creating *rag2:myr-mAkt2* transgenic fish, Gutierrez et al. also showed mAkt2 by itself could induce T-ALL, albeit less potently than *hMYC* [55]. In *mAkt2*-transgenic controls, 17% of fish developed T-ALL by 20 weeks, demonstrating Akt alone could induce T-ALL. To our knowledge, this model has not been explored further, but could likely also advance our understanding of T-ALL molecular events [55].

#### 2.2.5. Model 5: Zebrafish NOTCH T-ALL Models

##### Model 5a: Transgenic Human NOTCH1–*Tg(rag2:hICN1-EGFP)* Zebrafish

NOTCH1-activating mutations occur in >50% of human T-ALL cases, across multiple molecular subtypes [58]. Hence, it was natural that one of the earliest zebrafish T-ALL models focused on this gene. Specifically*,* Chen et al. created transgenic zebrafish expressing “human intracellular NOTCH1” (*hICN1*), a constitutively active truncated NOTCH1 protein [59]. As in some *mMyc* models, *hICN1* was fused to EGFP, and the zebrafish *rag2* promoter was once again used. T-ALL developed in >40% of mosaic animals by five months, sufficient latency to maintain the line. ALL were confirmed to be T-lineage, oligoclonal, and transplantable. As in *mMyc* and *hMYC* fish, T-ALL were aggressive, invading throughout animals. By RT-PCR, the authors showed hICN1 activated *D. rerio* homologues of known human NOTCH1 targets such as *her6* and *her9*. Unfortunately, stable *rag2:ICN1-EGFP* transgenic fish showed longer latency (~11 months; penetrance remained ~40%), and likely due to this, this model has not been widely studied.

• Key Discoveries

When *hICN1*-transgenic fish were crossed to animals overexpressing anti-apoptotic zebrafish *bcl2*, T-ALL showed accelerated onset and higher incidence, indicating synergy between NOTCH1 and BCL2. Progression from T-LBL to T-ALL was not reported upon, but in view of later findings from double-transgenic *mMyc* + *bcl2* fish [48], such investigations are perhaps warranted. Notably, in multiple studies, endogenous *myca* and *mycb* are not up-regulated in ICN1-induced T-ALL [59,60], although NOTCH1 induces MYC in human T-ALL [34,36,61]. He et al. proposed that *D. rerio* c-MYC homologues may lack upstream NOTCH1-driven enhancers, which are described in mammals, potentially explaining this difference [13,62]. This apparent distinction allows investigators to examine NOTCH1- and MYC-mediated actions separately in *D. rerio* T-ALL, which is not possible in mammals, but also highlights potentially important mechanistic differences between zebrafish and human T-ALL.

##### Model 5b: Transgenic Zebrafish Notch1–*Tg (rag2:znotch1a ^ICD^)*

Blackburn et al. created fish with either *rag2:hICN1* or *rag2:znotch1a^ICD^* (*notch1a* is one of two zebrafish *NOTCH1* homologues; “ICD” refers to a truncation analogous to human ICN1). Both were paired with *rag2:mMyc* in double-transgenic animals [60]. As noted, zebrafish are well-suited to study independent roles of NOTCH1 and MYC, since unlike in mammals, hICN1 does not up-regulate endogenous zebrafish *myc* homologues. Moreover, to date, *notch1a* and *notch1b* mutations have not been reported in zebrafish MYC-driven T-ALL, despite the high prevalence of *NOCTH1* mutations in human T-ALL. In this report, *hICN1*/*znotch1a^ICD^* augmented *mMyc* to accelerate cancer onset, but did not alter T-ALL proliferation or apoptosis. RNA expression studies also revealed a shared signature in fish, mouse, and human T-ALL, suggesting conservation of key molecular circuitry for a ‘Notch profile.’ They hypothesized that NOTCH1 mutations are likely an initiating event in human T-ALL that expands pre-malignant thymocytes, with only a subset of cells acquiring additional mutations to become fully transformed.

• Key Discoveries

NOTCH1 activation alone is insufficient to induce T-ALL, requiring additional oncogene-activating and/or tumor suppressor-inactivating mutations. At least in zebrafish, NOTCH and MYC do not collaborate to enhance LIC, as T-ALL with mMyc *and* Notch1a^ICD^ or those with only mMyc show similar LIC frequencies [60].

#### 2.2.6. Models 6–8: Non-Transgenic T-ALL Models

Frazer et al. reported a chemical mutagenesis forward-genetic screen for T-ALL-prone zebrafish [63]. They exposed *lck:EGFP* transgenic fish, which express GFP highly in T-lineage cells due to their ardent *lck* expression [64], to N-ethyl-N-nitrourea (ENU), an alkylator that randomly induces genomic mutations. ENU-exposed males then sired large clutches of fish that were screened for GFP+ tumors. To verify heritable transmission, progeny of fish developing T-ALL were likewise screened for GFP+ cancers. Ultimately, they isolated three cancer-predisposed *D. rerio* lines, dubbed *hlk, srk,* and *otg*, and demonstrated malignancies were T-ALL. These lines exhibited different T-ALL penetrance in homozygous and heterozygous mutants, with peak incidence at 5–9 months, and unique expression signatures in non-malignant T cells and T-ALL of each mutant. Morphologic features of T-ALL and patterns of disease spread resembled transgenic MYC and NOTCH zebrafish, as well as in human T-ALL. TCRβ analyses confirmed T-ALL clonality, and allogeneic transplants verified transformed T cells were immortal. Unfortunately, due to incomplete penetrance in both homozygous vs. heterozygous mutants and the highly polymorphic nature of the zebrafish genome, the germline mutations that *hlk*, *srk*, and *otg* represent are unidentified.

##### Key Discoveries

Overall, this report was unique in proving that endogenous mutations—as opposed to transgenes—could also drive zebrafish T-ALL. It further established that forward-genetic approaches could successfully identify complex phenotypes like cancer-predisposition that manifest later in development (i.e., not in embryos/larvae). These models were subsequently used to explore somatically acquired genetic changes occurring in T-ALL, specifically those contributing to relapse. Rudner et al. serially allo-transplanted *hlk*, *srk*, and *otg* T-ALL, examining primary and passaged cancers via array comparative genomic hybridization (aCGH). [65] Copy number aberrations were then compared to those in a cohort of 75 human T-ALL also analyzed by aCGH. This revealed several amplified and deleted genes shared by T-ALL of both species, including loci linked to poor clinical outcome.

#### 2.2.7. Model 9: T-ALL Induced by ARID5B–*Tg(rag2:hARID5B)* Zebrafish

Leong et al. identified roles for AT-rich interactive domain 5B (ARID5B) in T-ALL expression patterns, cell growth, and survival. [66] Specifically, they showed ARID5B reinforced an oncogenic signature that positively regulated both MYC and a TAL1/SCL. ARID5B also regulated other members of the TAL1 transcriptional complex besides TAL1, and co-occupied TAL1 targets with TAL1, reinforcing a genetic program that drives T-ALL. Reciprocally, they showed ARID5B is a TAL1 transcriptional target. Employing a familiar strategy, they built *rag2:hARID5B* + *rag2:mCherry* double-transgenic zebrafish to analyze ARID5B overexpression, and this stable transgenic line displayed several interesting phenotypes: (1) Thymic involution, which normally begins at ~3 months, was delayed (and perhaps prevented). (2) Non-malignant thymocytes showed marked γ-irradiation resistance. (3) A small number of fish (2/38; ~5%) developed T-LBL/ALL by six months, with cancers overexpressing either endogenous *myca* or *mycb*, and other genes consistent with an immature thymocyte developmental arrest.

##### Key Discoveries

This line had low T-ALL penetrance, which may limit its future applications. Likewise, characterization was limited to analysis of only two T-ALL. Nonetheless, this study provides an example of how zebrafish can rapidly interrogate candidate genes in vivo. Here, the preponderance of work was performed in Jurkat and other human T-ALL cell lines, implicating ARID5B as a possible T-ALL oncogene. To functionally test this, the authors built a straightforward model, and using simple fluorescent microscopy assays (monitoring thymic size with aging, after irradiation, or to identify tumors) were able to demonstrate that ARID5B can in fact induce T-ALL.

#### 2.2.8. Model 10: T-ALL Induced by jdp2–*Tg(rag2:zjdp2)* Zebrafish

Mansour et al. studied JDP2, a transcription factor implicated by unbiased genome-wide T-ALL studies in mice, using zebrafish [67]. In humans, they found JDP2 expression was normally restricted in developing thymocytes, but high in many T-ALL, particularly the early thymocyte progenitor subtype, ETP-ALL. T-ALL cases with higher JDP2 expression had inferior outcomes, especially ETP-ALL patients. JDP2 depletion caused apoptosis of human T-ALL cell lines, and they linked this transcriptional regulation of anti-apoptotic MCL1 by JDP2. They also performed several zebrafish studies: (1) Fish injected with *rag2:mMyc* + *rag2:zjdp2* showed accelerated T-ALL onset compared to fish with *rag2:mMyc* alone. (2) Stably transgenic *rag2:zjdp2* fish exhibited thymic hyperplasia, delayed involution, and after 9–12 month latency, ~50% incidence of T-ALL that could engraft into immunodeficient larvae and adult fish. (3) Supporting their prior results, *rag2:zjdp2* thymocytes had elevated *mcl1* RNA, and presumably due to this, were resistant to glucocorticoids. Overall, this study convincingly demonstrated oncogenic roles for JDP2 in T-ALL, as well as a link to MCL1 that may explain why JDP2+ T-ALL patients do poorly.

##### Key Discoveries

This report provided evidence that JDP2 is not only an oncogenic driver (proven by *rag2:zjdp2* fish developing T-ALL), but also a possible cause of treatment failure (revealed by thymocyte glucocorticoid resistance). Comparing steroid sensitivity of *mMyc*- vs. *zjdp2*-driven T-ALL could have addressed this even more definitively. Even so, models like *rag2:zjdp2* and *rag2:mMyc* + *rag2:zbcl2* fish provide powerful in vivo templates to discover and pre-clinically test novel agents that circumvent steroid resistance, an important problem not just in T-ALL, but lymphocyte malignancies in general, including B-ALL.

### 2.3. B-ALL: Introduction

B-ALL is the most common type of ALL in both children and adults, representing 85% of pediatric ALL and 75% of adult cases [68,69]. Like T-ALL, which are often classified by oncogenic transcription factors [61,70], B-ALL can be separated into molecular subgroups depending on their genetic aberrations [71]. Many subgroups are defined by translocations that create oncogenic fusion proteins that then transform B cell progenitors, such as the ”Philadelphia chromosome” (*BCR-ABL1*), *ETV6-RUNX1*, and *TCF3-PBX1* [68]. Other B-ALL subtypes are characterized by global genetic features, such as hyperdiploidy (>50 chromosomes) or hypodiploidy (<44 chromosomes) [68], or specific transcriptional profiles: *PAX5*-driven subtypes, *DUX4*-rearranged, *BCR-ABL1*-like*,* or *ETV6-RUNX1*-like [71,72,73]. Subtype frequencies differ between children and adults, but essentially all types occur in both patient populations [71]. Translocations or specific mutations are likely initiating events in B-ALL, but additional (epi) genetic events are probably required for B-ALL to develop [73]. Several murine models develop B-ALL when engineered with human B-ALL driver mutations, such as *MLL* fusions [74,75,76]. B-ALL incidence increases with the addition of secondary genetic events, like *Bcl2* overexpression or *Kras* activation [75]. Few reports of zebrafish B-ALL exist, although differences between *D. rerio* and human B cells are comparably similar to those between T cells of these species [77]. The many successful zebrafish T-ALL models suggest *D. rerio* can likely emulate several human B-ALL subtypes, but currently, zebrafish B-ALL models lag behind [13,15]. Here, we will review the three zebrafish B-ALL models reported to date (Table 1), but it is likely many others will soon follow.

### 2.4. B-ALL Zebrafish Models

#### 2.4.1. Model 1: Transgenic *hTEL-AML1* Zebrafish

The t(12;21)(p13;q22) translocation encodes a *TEL-AML1* (*ETV6-RUNX1*) fusion protein occurring in the most common pediatric B-ALL subtype, representing ~25% of cases [72,73]. To recapitulate this, Sabaawy et al. created a transgenic *hTEL-AML1* zebrafish line [78]. In this transgenic line, the human fusion *TEL-AML1* was expressed under the control of three different promoters: *Xenopus ef1a* (*Xef1a*), zebrafish *beta-actin* (*zba*), and zebrafish *rag2*. *Xef1a* and *zba* were used to test global expression of *TEL-AML1* and the *rag2* promoter was used to direct lymphoblast-specific expression. Somewhat surprisingly, global *TEL-AML1*, but not lymphoblast-specific expression, induced B-ALL in ~3% of fish after a ~12 month latency. Sabaawy et al. also reported ~6% of ubiquitously expressing *TEL-AML1* fish developed fatal lymphoid hyperplasia as early as 28 dpf with manual differential blood cell counts confirming increased numbers of immature blast-like cells (~10%–15%, just below the level required to diagnose ALL in patients). A unique feature of this B-ALL compared to subsequent B-ALL models was its origin in the kidney-marrow, which was confirmed by successful transplantation of kidney-marrow cells of *Xef1a:TEL-AML1* leukemic fish into irradiated recipients, where all transplanted recipients developed B-ALL 6–9 weeks after transplant.

##### Key Discoveries

• B-ALL induced by Global, but not by Lymphoblast-Specific, *TEL-AML1* Expression

Sabaawy et al. showed only global expression of *TEL-AML1* could induce B-ALL, with low incidence and long latency. They hypothesized that B-ALL did not occur in *rag2*:*hTEL-AML1* fish because TEL-AML1 was needed in less-mature stem/progenitor cells that do not yet express *rag2*. In fish with global expression of TEL-AML1, gene expression profiles showed deregulation of anti-apoptotic genes, such as *bcl2, bcl-xl,* and *bax,* which inhibit apoptosis and/or promote cell cycle arrest in HSC.

• *TEL-AML1* induced Zebrafish B-ALL Resembles Human *CD10*^+^ pre-B-ALL

B-ALL in *TEL-AML1* transgenic zebrafish were negative for *tcr-a* and *igm,* but positive for *cd10*, *ikaros* (a transcription factor expressed by lymphoid progenitors), and *tal1* (a stem cell marker), which is similar to human *CD10*^+^ pre-B-ALL [78]. A drawback of this model is its long latency and low incidence, however, these characteristics are also features of human *TEL-AML1* B-ALL [79], arguably making this model a faithful reproduction of its human counterpart. Global *TEL-AML1-*expressing zebrafish B-ALL showed significant down-regulation of endogenous zebrafish *tel*. This may explain the long latency and low incidence of B-ALL in these fish, as two *tel* alleles would need to be disabled, rather than one (in human patients, the first *TEL* is disabled by the translocation itself). The expression signature they observed suggests that *TEL-AML1* activity in uncommitted progenitors, but not common lymphoid progenitors, induces pre-leukemic clones that are arrested at pre-B cell stages. Then, additional acquired genetic events may allow pre-leukemic clones to proliferate and/or block apoptosis, leading to outright ALL. This sequence of events is consistent with those in *TEL-AML1* B-ALL patients, where *TEL-AML1*-expressing progenitor cells expand prenatally as a pre-leukemic population. This provides a window for further postnatal genetic events to convert pre-leukemic progenitors into overt leukemia [80,81]. Due to the specificity of this model, it is ideally suited to study the cellular origin and the molecular pathogenesis of *TEL-AML1*-induced B-ALL, which represents >20% of pediatric B-ALL [71].

#### 2.4.2. Model 2: B-ALL in *rag2:mMyc* Transgenic Zebrafish

MYC is hyperactive in many B cell cancers [82] and *MYC* translocations occur in up to 7% of B-ALL [71,83,84]. In addition, B-ALL express *MYC* at high levels [54], and *MYC* overexpression is linked to relapse and refractory disease in adult B-ALL [85]. Thus, it might be expected that *rag2:mMyc* zebrafish would develop B-ALL, since B-lymphoblasts express *rag2*, and MYC is known to drive human B-ALL. However, although this line was first described in 2003 [32], it was only recently realized that they also develop B-ALL [86]. As noted previously, zebrafish T-ALL can be passaged by allo-transplant, facilitating study of monoclonal ALL [50]. Such studies first uncovered clonal heterogeneity in T-ALL, and then recently, distinct ALL subtypes in *rag2*:*mMyc* fish: T-ALL with short latency and high LSC frequency, and previously unrecognized B-ALL with longer latency and less LSC [50,86]. These latter cancers expressed homologues of human B-lymphoblast genes, such as *rag1 and rag2*, *pax5*, *dntt* (a *TdT* orthologue), and *cd79a* (aka, *Ig*). In addition to known B cell markers, *mMyc* B-ALL also expressed *gfi1ab*, *zfx3*, *notch1a*, *nf1b*, and *gtf3aa*; analysis of this signature suggested up-regulation of ribosome biogenesis, RNA binding, and other biologic pathways [53].

##### Key Discoveries

*mMyc* B-ALL expressed *ighm* constant regions, but transcripts that included recombined immunoglobulin (Ig) variable regions were not detected. Constant region mRNAs may represent ‘sterile transcripts,’ which mammalian B cells express during V(D)J and class switch recombination (CSR) [87,88,89]. The authors interpreted these data to mean that *mMyc* B-ALL arrest at an early pro-B cell stage, prior to Ig heavy chain VDJ rearrangement [53]. The apparently lower incidence of B-ALL in *mMyc* fish may impede studying these further, but B-ALL incidence in this line is not yet reported, so this is currently unclear. Even so, this model can potentially reveal how MYC hyperactivity transforms pro-B cells.

#### 2.4.3. Model 3: B-ALL in *rag2:hMYC-ER* Transgenic Zebrafish

Like *rag2*:*mMyc* fish, *rag2*:*hMYC-ER* fish were known to develop T-ALL [55]. However, Borga et al. recently reported B-ALL in the *rag2*:*hMYC-ER* line. They built a double-transgenic with *hMYC* in addition to an *lck*:*EGFP* marker transgene. Fortuitously, this marker is highly expressed by T cells, but lowly expressed by some—but not all—B cells [90]. Consequently, in *rag2:MYC-ER*;*lck:EGFP fish*, T-ALL are GFP^hi^, while B-ALL are GFP^lo^. This low *lck* expression by *hMYC* zebrafish B-ALL is consistent with low levels of *lck* in many human pre-B ALL [54]. Remarkably, many animals developed simultaneous GFP^hi^ and GFP^lo^ T- and B-ALL [54]. Histology of *hMYC* B-ALL demonstrated highly infiltrative disease that was indistinguishable from T-ALL (including thymic tumors), but expression patterns that were distinct from T-ALL and resembled human B-ALL [54]. *hMYC* B-ALL showed pre-B cell gene expression, such as *igic1s1*, *rag2*, *pax5*, and *cd79b* [54]. Borga et al. also transplanted *hMYC* GFP^lo^ B-ALL into irradiated immunosuppressed WT host fish to prove the ’immortality’ *hMYC* B-ALL, and transplanted B-ALL remained GFP^lo^ and stably expressed only B-lineage markers. Borga et al. also proved that *hMYC* B-ALL exhibited lower LSC frequencies compared to *hMYC* T-ALL, mirroring *mMyc* fish. One interesting question raised by both the *hMYC* and *mMyc* models is why B-LSC are lower frequency than their T-LSC counterparts.

##### Key discoveries

To date, the Borga et al. paper is the only large study of *D. rerio* B-ALL, as the *TEL-AML1* and *mMyc* papers presented detailed results from only five and two B-ALL, respectively. However, even with this caveat, intriguing differences between *hMYC*- and *mMyc*-driven B-ALL have already been recognized, suggesting these models are not interchangeable [53]. As noted above, *mMyc* B-ALL expressed IgM constant region transcripts. Curiously, *hMYC* B-ALL did not express *ighm*, but rather *ighz*, which encodes an isotype specific to fish, IgZ [53,54]. In zebrafish, isotype switching by CSR has not been described; instead, B cells ”choose” between IgM and IgZ during V(D)J recombination, resulting in distinct B cell lineages [53]. Consistent with the notion of ALL in different B cell lineages, *hMYC* and *mMyc* B-ALL displayed surprisingly disparate expression patterns and pathway signatures, suggesting the models are complementary. The reason(s) that transgenic *mMyc* and *hMYC* (which encode 92% identical proteins) act differently in *D. rerio* B-lymphoblasts remain to be elucidated, but MYC’s importance in human B-ALL warrants further investigation of this phenomenon [85,91]. The potential existence of *hMYC-*driven GFP^neg^ B-ALL also remains unclear, and it is possible *hMYC* induces B-ALL in both the IgZ- and IgM-lineages. GFP^neg^ B cells were detected in this model, [54] but these *lck* negative B cells have not yet been thoroughly characterized. Using alternative fluorescent reporter lines, such as *cd79a*:*EGFP* or *cd79b*:*EGFP* in place of *lck*:*EGFP* [92], the possibility of other B-ALL types in the *hMYC* model can be definitively addressed.

## 3. Conclusions

This review has highlighted published zebrafish ALL models that have contributed to our current understanding of key players and pathways in human ALL. However, *D. rerio* ALL models continue to emerge, with newer T- and B-ALL transgenic lines reported at recent conferences that will soon join the existing literature. Together, these vertebrate ALL models can teach us about mechanisms of ALL progression and resistance and provide platforms to screen and pre-clinically test novel pharmacologic agents. As noted throughout this review, many findings in zebrafish have been recapitulated or confirmed in human ALL samples and cell lines, or vice versa. In an era where the genetic revolution and oncology intersect ever-more-frequently, expedient and cost-effective models to survey gene function in vivo are at a premium. Going forward, zebrafish models of ALL will continue to allow us to work towards conquering this devastating disease—one fish at a time.

## Figures and Tables

**Table 1 ijms-20-05313-t001:** Summary of zebrafish acute lymphoblastic leukemia (ALL) models.

T-ALL Models		Transgenes (Promoter:Oncogene)			Incidence (%)	Mean Latency (dpf)	Key Features	Other Remarks
		Promoter	Oncogene(s)	Full Construct				
Model 1	Murine *Myc* (*mMyc*)	*rag2*	*mMyc*	*rag2:EGFP-mMyc*	100	52	T-ALL mirrored most common TAL1 human T-ALL subtype; used as a foundation for several ensuing T-ALL studies	First transgenic zebrafish ALL and cancer model; complete penetrance and short latency created challenges [32]
Model 2a	*Cre*-*Lox* conditional *mMyc*	*rag2*	*mMyc*	*rag2:loxP-dsRED2-loxP-EGFP-mMyc*	7	151	Used *Cre* recombinase injection to induce *EGFP*-*mMyc* expression	Low penetrance [40]
Model 2b	Heat shock inducible *Cre* with *mMyc*	*rag2*	*mMyc*	*hsp70:Cre;rag2:LDL-EGFP-mMyc*	80	120	Used heat shock-inducible *Cre* transgene to markedly improve T-ALL penetrance	Easier to maintain and amenable to forward-genetic screens [46]
Model 2c	Above line with *bcl2*	*rag2*	*mMYC/bcl2*	*hsp70:Cre;rag2:LDL-EGFP-mMyc;rag2:EGFP-bcl2*	>80	<120	T-LBL favored over T-ALL due to constitutive *bcl2* expression	Used to study differences between T-LBL and T-ALL [48]
Model 3	Co-injection of multiple transgenes	*rag2*	*mMyc*	*rag2:mMyc and rag2:dsRED2 co-injected into stably transgenic rag2:GFP fish*	N/A	N/A	Proved use of co-injection, allowing hundreds of embryos to be injected with multiple transgene combinations	Used to color-code T-ALL in the syngeneic CG1 background for LIC and cancer heterogeneity studies [49]
Model 4	Human *MYC* (*hMYC*)	*rag2*	*hMYC*	*rag2:hMYC-ER*	100 (4HT treated fish)	37	4HT-inducible *hMYC* expressing T-ALL used to study MYC and PI3K-AKT interaction.	Showed BIM repression is a key event downstream of MYC and PI3K-AKT in resistant T-ALL; also used to show therapeutic potential for PP2A in T-ALL [55]
	Myristoylated murine *Akt2* (*myr-mAkt2*)	*rag2*	*myr-mAkt2*	*rag2:myr-mAkt2*	17	140	Showed *myr-mAkt2* could induce T-ALL in isolation	Low incidence and has not been extensively studied [55]
Model 5a	Human *NOTCH1* (*hICN1*)	*rag2*	*hICN1*	*rag2:hICN1-EGFP*	40	330	*NOTCH1* induced T-ALL without up-regulating endogenous *myca* and *mycb*	Allows NOTCH1 and MYC to be studied separately, not possible in mammals [59]
Model 5b	Zebrafish *notch1a* (*znotch1a*)	*rag2*	*znotch1a^ICD^*	*rag2:znotch1a^ICD^*	100	<52	hICN1/znotch1a^ICD^ accelerated T-ALL, but did not alter T-ALL proliferation or apoptosis	Showed NOTCH1 activation alone was insufficient to induce T-ALL [60]
Model 6–8	Germline mutants: *hlk, srk,* and *otg*	N/A	N/A	N/A	40–50 in homozygotes	150–270	Showed endogenous mutations could also drive zebrafish T-ALL	Mutant genes were never identified [63]
Model 9	Human *ARID5B*	*rag2*	hARID5B	*rag2:hARID5B*	5	180	Proved role of *ARID5B* in T-ALL	Fish also showed delayed thymic involution, and thymocytes showed radiation resistance. [66]
Model 10	Zebrafish *jdp2*	*rag2*	*zjdp2*	*rag2:zjdp2*	50	270–360	Proved role of *jdp2* in T-ALL and linked *mcl1* to drug resistance	Fish also showed thymic hyperplasia and delayed thymic involution [67]
B-ALL Models		Transgenes (promoter:oncogene)			Incidence	Mean Latency (dpf)	Key Features	Other Remarks
		Promoter	Oncogene(s)	Full Construct				
Model 1	Human *TEL-AML1* (*ETV6-RUNX1*)	*Xenopus ef1a (Xef1a)/zebrafish beta-actin (zba)*	*TEL-AML1*	*Xef1a:TEL-AML1 and zba:TEL-AML1*	3	360	Long latency and low incidence recapitulate human *TEL-AML1* B-ALL	Models the most prevalent type of pediatric ALL [78]
Model 2	Murine *Myc* (*mMyc*)	*rag2*	*mMyc*	*rag2:EGFP-mMyc*	Not reported	Not reported	Develops *ighm+* B-ALL (in addition to T-ALL)	Unexpectedly unique expression signature from *hMYC* B-ALL [79]
Model 3	Human *MYC* (*hMYC*)	*rag2*	*hMYC*	*rag2:hMYC-ER*	Not reported	Not reported	Develops *ighz+* B-ALL (in addition to T-ALL)	Only large zebrafish B-ALL study; unexpectedly unique expression signature from *mMyc* B-ALL [54]

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
