# Peer review of "Tackling Acute Lymphoblastic Leukemia—One Fish at a Time"

_ijms, 2019, doi:10.3390/ijms20215313_

Round 1

Reviewer 1 Report

The authors present a comprehensive review of zebrafish models for ALL.

Although there is no shortage of reviews on zebrafish models of leukemia, this one focuses specifically on ALL. It describes and reviews each of the models so far, including strengths and limitations with a summary of the key discoveries generated.

This is a clear, well-written, well-organized review, which will be useful not only to those interested in using zebrafish to model leukemia but also to those looking to model other malignancies, as it lays out some of the potential pitfalls of this kind of modelling as well as a compendium of the kinds of approaches that are successful.

Minor edits:

Title page: I believe the author should be Frazer (with a final “r”)

Graphical abstract

Generally works well

Some of the text is stretched (eg transgenesis) and this should be corrected before publication

Suggest replacing “ease of use” with “in vivo visualization”

6000 deaths should include per year (if not in the graphic, in the legend)

Left hand arrow could point more directly to circle with ALL crossed out

Line 133: ”even in zebrafish”; suggest “across species”

Line 155: delete “of”

Line 189: occurred [in] 13%

Line 214: delete “ed”; should be “levels allowed”

Line 256: latency [and] increase

Line 278 and elsewhere: myristolated should be myristoylated

Lines 278-9: comma after “pten” and “transgene” to clarify meaning.

Line 339: delete “are” at end of line

Line 341: fully transformed (delete hyphen)

Line 502: “theser” – perhaps “these further”?

Table:

The table would benefit from a references column or from including references in the “other remarks” column.

There was a formatting issue in the reviewer document that split words in Models and Oncogene columns. To be corrected in proof, if necessary.

Author Response

October 21, 2019

Dear Drs. Chiarini and Dovat,

We are submitting a revision of manuscript ijms-623664, entitled “Tackling Acute Lymphoblastic Leukemia – One fish at a time” for publication as a review article in the special Issue “Molecular Research on Acute Lymphoblastic Leukemia” of International Journal of Molecular Sciences.” 

We appreciate the incredibly thorough review and fully agree with all comments. We have made all changes as suggested and believe this has improved the manuscript. Please see our point by point response below.

Sincerely,

Arpan Sinha MD and

Kimble Frazer MD, Ph.D.

Response to Reviewer 1 Comments

Minor edits:

Title page: I believe the author should be Frazer (with a final “r”)

Graphical abstract

Generally works well

Some of the text is stretched (eg transgenesis) and this should be corrected before publication

Suggest replacing “ease of use” with “in vivo visualization”

6000 deaths should include per year (if not in the graphic, in the legend)

Left hand arrow could point more directly to circle with ALL crossed out

Line 133: ”even in zebrafish”; suggest “across species”

Line 155: delete “of”

Line 189: occurred [in] 13%

Line 214: delete “ed”; should be “levels allowed”

Line 256: latency [and] increase

Line 278 and elsewhere: myristolated should be myristoylated

Lines 278-9: comma after “pten” and “transgene” to clarify meaning.

Line 339: delete “are” at end of line

Line 341: fully transformed (delete hyphen)

Line 502: “theser” – perhaps “these further”?

Table:

The table would benefit from a references column or from including references in the “other remarks” column.

There was a formatting issue in the reviewer document that split words in Models and Oncogene columns. To be corrected in proof, if necessary.

Response 1: Thank you for your thorough review. We appreciate the comments. The edits to the title page, graphical abstract, table and manuscript have been made. In addition, we have also corrected the headings for each model to be uniformly italicized.

Reviewer 2 Report

The authors clearly describe several zebrafish T-ALL models and newer zebrafish B-ALL models, summarize the key discoveries highlighted by the models and discuss their potential utility for drug discovery. The review is well written and updated and two of the authors have contributed original studies in the field.

Author Response

October 21, 2019

Dear Drs. Chiarini and Dovat,

We are submitting a revision of manuscript ijms-623664, entitled “Tackling Acute Lymphoblastic Leukemia – One fish at a time” for publication as a review article in the special Issue “Molecular Research on Acute Lymphoblastic Leukemia” of International Journal of Molecular Sciences.” 

We appreciate the thoughtful review by the reviewers. We have carefully addressed the reviewer’s concerns and believe this has improved the manuscript. Please see our point by point response below.

Sincerely,

Arpan Sinha MD and

Kimble Frazer MD, Ph.D.

Response to Reviewer 2 Comments

Comments and Suggestions for Authors

The authors clearly describe several zebrafish T-ALL models and newer zebrafish B-ALL models, summarize the key discoveries highlighted by the models and discuss their potential utility for drug discovery. The review is well written and updated and two of the authors have contributed original studies in the field.

Response 1: Thank you for your review. We appreciate the comments. Minor edits for correction of typos has been made to the title page, graphical abstract and the manuscript. In addition, we have also corrected the headings for each model to be uniformly italicized.